# Hyperimmunized Chickens Produce Neutralizing Antibodies against SARS-CoV-2

**DOI:** 10.3390/v14071510

**Published:** 2022-07-09

**Authors:** Emily J. Aston, Michael G. Wallach, Aarthi Narayanan, Sofia Egaña-Labrin, Rodrigo A. Gallardo

**Affiliations:** 1Department of Animal Science, College of Agricultural and Environmental Sciences, University of California-Davis, Davis, CA 95616, USA; aston.emily@gmail.com; 2School of Life Sciences, Faculty of Science, University of Technology Sydney, Sydney, NSW 2007, Australia; michael.wallach@uts.edu.au; 3National Center for Biodefense and Infectious Diseases, George Mason University, Fairfax, VA 22030, USA; anaraya1@gmu.edu; 4Department of Population Health and Reproduction, School of Veterinary Medicine, University of California-Davis, Davis, CA 95616, USA; seganal@ucdavis.edu

**Keywords:** COVID-19, SARS-CoV-2, chicken, passive immunization, antibodies, neutralizing antibodies, egg

## Abstract

The novel severe acute respiratory syndrome (SARS) coronavirus, SARS-CoV-2, is responsible for the global COVID-19 pandemic. Effective interventions are urgently needed to mitigate the effects of COVID-19 and likely require multiple strategies. Egg-extracted antibody therapies are a low-cost and scalable strategy to protect at-risk individuals from SARS-CoV-2 infection. Commercial laying hens were hyperimmunized against the SARS-CoV-2 S1 protein using three different S1 recombinant proteins and three different doses. Sera and egg yolk were collected at three and six weeks after the second immunization for enzyme-linked immunosorbent assay and plaque-reduction neutralization assay to determine antigen-specific antibody titers and neutralizing antibody titers, respectively. In this study we demonstrate that hens hyperimmunized against the SARS-CoV-2 recombinant S1 and receptor binding domain (RBD) proteins produced neutralizing antibodies against SARS-CoV-2. We further demonstrate that antibody production was dependent on the dose and type of antigen administered. Our data suggests that antibodies purified from the egg yolk of hyperimmunized hens can be used as immunoprophylaxis in humans at risk of exposure to SARS-CoV-2.

## 1. Introduction

The novel severe acute respiratory syndrome (SARS) coronavirus, SARS-CoV-2, is the etiologic agent of COVID-19, a newly emerged viral respiratory disease in humans. First identified in late 2019, the disease reached pandemic status and continues to be a significant public health threat worldwide. The COVID-19 landscape is rapidly evolving, and effective strategies to mitigate its spread are urgently needed. Notably, the first emergency use authorizations for a vaccine for the prevention of COVID-19 were issued on 11 December 2020 [1]. Multiple vaccines and treatments have since been developed and approved for emergency use. Although vaccines appear reasonably effective at reducing the severity of illness associated with COVID-19, the recent emergence of new variants threatens the efficacy of these vaccines. Therefore, ongoing research to develop new vaccines, antiviral drugs, and antibody therapies is critical to reducing the devastating impact of the pandemic. Successful COVID-19 intervention likely requires multiple approaches, and antibody therapies are an attractive strategy to protect at-risk individuals from SARS-CoV-2 infection.

The genetic material of coronaviruses is composed of a single-stranded, positive-sense RNA genome [2]. This genome is the largest of the RNA viruses and ranges from 27 to 30 kilobase pairs [2]. SARS-CoV-2 is a betacoronavirus belonging to the family *Coronaviridae* [2]. Coronaviruses cause respiratory and gastrointestinal disease in a wide range of animal species, including mammals and birds [2]. Alphacoronaviruses and betacoronaviruses are known to infect humans and other mammals, and gammacoronaviruses and deltacoronaviruses infect birds, although some can infect mammals [2]. The most notable and extensively studied gammacoronavirus in chickens is infectious bronchitis virus (IBV), against which most commercial hens are immunized [2].

Coronavirus entry into cells is mediated by the spike protein, most specifically its S1 portion, a peplomer-like structure anchored to the virus membrane in the form of a trimer. In addition, the S1 subunit of this protein determines virus variability and elicits neutralizing antibodies [2,3,4]. On the globular head of each SARS-CoV-2 S1 protein is a receptor-binding domain (RBD), which specifically recognizes the human angiotensin-converting enzyme 2 (ACE2) [5]. The RBD is the most antigenic region of the spike protein in coronaviruses and is thus an attractive site for therapies that target cell entry of the virus.

One such therapy was approved by the U.S. Food and Drug Administration on 23 August 2020 for an emergency use authorization for investigational convalescent plasma to treat hospitalized COVID-19 patients [6]. The principle behind this therapy is that recovering COVID-19-infected individuals develop SARS-CoV-2-specific antibodies that can be recovered from plasma and administered to ill patients to neutralize the virus if applied systemically. This passive transfer of antibodies is not limited only to SARS-CoV-2-infected patients. Researchers are also applying this concept of virus neutralization to generate SARS-CoV-2-specific antibodies in animals to be used for passive immunization against the virus in humans [7]. Harvesting antibodies from eggs laid by hens that have been immunized against the spike protein of SARS-CoV-2 is an attractive model to produce protective antibodies due to the scalability, convenience, and low cost [7].

Chickens produce immunoglobulin Y (IgY), which is a homologue of mammalian IgG. An average egg yolk yields 50–100 mg of IgY, of which 2–10% comprise specific antibodies [8,9]. When hyperimmunizing hens, the amount of antigen-specific IgY produced will depend on the age of the hen, adjuvant, route of application, as well as the dose, antigenicity, and molecular weight of antigen administered to each hen [10,11,12,13]. A laying hen lays an average of 300 eggs per year, which corresponds to approximately 15–30 g of IgY. Thus, a considerable amount of polyclonal antibody can be non-invasively recovered from the eggs laid by immunized chickens. Other advantages of using IgY in human applications are that it is well tolerated and can be administered orally [14,15]. Furthermore, IgY neither binds to human rheumatoid factors nor activates the human complement system [14]; therefore, reducing the risk of inflammatory reactions as a secondary effect of using antibodies produced in different species. These characteristics make chicken IgY a promising source of new therapies for human viral diseases such as COVID-19 in addition to vaccination strategies [16].

Here, we demonstrate that hens hyperimmunized against the SARS-CoV-2 recombinant receptor binding domain and/or S1 protein produced neutralizing antibodies against SARS-CoV-2. By hyperimminunizing and testing for antibodies against SARS-CoV-2 via enzyme-linked immunosorbent assay (ELISA) and plaque-reduction neutralization assay (PRNA), we further demonstrate that antibody production was dependent on the dose and type of antigen administered. Our data suggests that antibodies purified from the egg yolk of hyperimmunized chickens can be used as immunoprophylaxis in humans at risk of exposure to SARS-CoV-2.

## 2. Materials and Methods

### 2.1. Constructs and Virus

The constructs used for vaccines A and B included the SARS-CoV-2 S1-derived RBD protein #0570, which included amino acids 328-533 and a His tag (vaccine A) and SARS-CoV S1-derived RBD amino acids 319-542 and a His tag (vaccine B), using a bacterial based in vitro expression system [17]. The reference virus used to obtain these constructs was the Wuhan strain. The construct used for vaccine C consisted of the SARS-CoV-2 S1 protein, including amino acids 16-685 and an Fc-tag, which was expressed in HEK293 cells (Cat # S1N-C5255, AcroBiosystems, Newark, DE, USA); these proteins were glycosylated.

Virus neutralization experiments were performed using the 2019-nCoV/USA-WA1/2020 strain sourced from an infected patient in Washington state.

### 2.2. Experimental Design

Two hundred 85-week-old laying hens were transported from a table egg layer farm in the California Central Valley to the UC Davis Teaching and Research Animal Care Services facility, where they were placed on pine shavings in climate-controlled BSL-2 rooms. After one week of acclimatization, the hens received two immunizations administered twelve days apart. Vaccines were prepared as an oil–water emulsion with an equal volume of Freund’s incomplete adjuvant (Thermo Scientific, IL, USA) using an Ultra-Turrax T25 High-Speed Homogenizer (IKA, Staufen, Germany) at 25,000 rpm for 10 min. The experimental groups were as follows: vaccine A twice (A/A), vaccine B twice (B/B), vaccine C twice (C/C), vaccine C followed by vaccine B (C/B), and adjuvant only (negative control). Each treatment group was divided into the following subgroups receiving a 2.5, 5, or 50 μg dose. All vaccines were administered by intramuscular route in the pectoral muscle in 0.5 mL volumes. Blood was collected from the ulnar vein upon arrival, 21 days following the second immunization, and at the end of the experiment (6 weeks post second dose). Blood was centrifuged and serum was stored at 4 °C for ELISA. One month following the second immunization, eggs were collected every week and stored at 4 °C for IgY extraction. In the sixth week after the last immunization, the hens were humanely euthanized by CO_2_ inhalation, immediately followed by cervical dislocation. A schematic of the experimental design is in Figure 1.

### 2.3. IgY Extraction from Egg Yolk

IgY extraction was performed as described previously [18], with minor modifications. Nine eggs per group were cracked open. The egg white was discarded, and the egg yolk was washed with phosphate-buffered saline (PBS). The egg yolk was punctured, and the vitelline membrane was discarded. The yolk was transferred to a 50 mL sterile centrifuge tube and Milli-Q^®^ water (Millipore purification system, Bedford, MA, USA) was added for a yolk dilution of 1:5. The tube was vortexed vigorously and stored overnight at 4 °C. The following day, the tubes were centrifuged at 1000× *g* for 30 min at room temperature. Since the supernatant was still yellow, a second dilution/centrifugation step was performed. Ten mL of the supernatant was collected and mixed with 40 mL of Milli-Q^®^ water for a final yolk dilution of 1:10, and centrifugation was performed at 2000× *g* at 15 °C for 30 min. The supernatant, now clear, was obtained and filtered through a Millex -HV 0.45 μm filter (Millipore, Bedford, MA, USA) and stored at 4 °C. To measure the IgY extraction efficiency, protein concentration of the yolk extract from 9 egg yolks was individually measured and compared with protein concentration from 10 individual serum samples obtained at necropsy from vaccine C group at 50 μg using the Take 3 plate on an Epoch Microplate Spectrophotometer (BioTek Instruments, Inc., Winooski, VT, USA).

### 2.4. IgY Antibody Titers

Serum and yolk IgY titers were measured by enzyme-linked immunosorbent assay (ELISA). Wells were coated with 0.5 μg of antigens A, B, or C in coating buffer and incubated at 4 °C overnight. All wash steps were performed with PBS-Tween 20 (0.05% Tween 20). Plates were blocked with 5% milk in PBS-Tween 20 for 1 h at room temperature. Serum samples were diluted 1:3200 in Sample Diluent, provided by the manufacturer, and purified yolk samples were diluted 1:400 in Milli-Q^®^ water. The rest of the ELISA was performed using the reagents provided by the commercial IDEXX infectious bronchitis Ab Test (IDEXX, Westbrook, ME, USA), according to the manufacturer’s instructions. Data were presented as sample to positive (S/P) ratio, which was calculated from the optical density (O/D) values using the following equation: (mean of test sample—mean of negative control)/(mean of positive control—mean of negative control). Three positive controls (A, B, and C) were prepared as a reference standard for each antigen. Each positive control was derived from sera from a hen immunized twice with 50 μg of one of the corresponding vaccines (A, B, or C) and selected for high IgY titer. Positive control sera were diluted in two-fold steps from 1:25 to 1:12,800 with the manufacturer-provided dilution buffer and incubated in wells coated with either 0.5 or 1 μg of corresponding antigen (A, B, or C). The final serum dilution and antigen coating concentration were determined by plotting the absorbance against the serum dilution and selecting the dilutions associated with the highest absorbance before saturation was observed. One negative control was used for all antigens. An S/P ratio allows titers to be compared across multiple plates, but S/P ratios should not be compared between antigens or between serum and egg yolk-derived IgY because a different positive control was used for each antigen and sample type.

### 2.5. Plaque Reduction Neutralization Assay

Antibodies derived from pooled sera and egg yolk from hens receiving vaccine A/A, B/B, C/C, and B/C at 50 μg dose were assessed for their ability to neutralize SARS-CoV-2 using a plaque reduction neutralization assay (PRNA). Briefly, the antibodies were diluted 1:10 in culture medium and incubated in two-fold serial dilutions with 100 plaque forming units (PFUs) of the Washington Strain of SARS-CoV-2 (USA-WA1/2020). SARS-CoV-2 was diluted in supplemented DMEM at a 1:1 ratio. The virus was then added to the antibody samples and allowed to incubate for 1 h at 37 °C and 5% CO_2_. After incubation, viral plaque assay was conducted to quantify viral titers. Twelve-well plates were previously seeded with Vero cells (ATCC CCL-81) at a density of 2 × 10^5^ cells per well. Media was aspirated from plates and virus-antibody samples were transferred to wells, one sample per well. Plates were incubated for 1 h at 37 °C and 5% CO_2_. After infection, a 1:1 overlay consisting of 0.6% agarose and 2× Eagle’s Minimum Essential Medium without phenol red, supplemented with 10% fetal bovine serum (FBS), non-essential amino acids (Gibco, 11140-050), 1 mM sodium pyruvate, 2 mM L-glutamine, and 1% P/S was added to each well. Plates were incubated at 37 °C for 48 h. Cells were fixed with 10% formaldehyde for 1 h at room temperature. Formaldehyde was aspirated and the agarose overlay was removed. Cells were stained with crystal violet (1% CV *w*/*v* in a 20% ethanol solution). Viral titer of SARS-CoV-2 was determined by counting the number of plaques and represented as plaque-forming units (PFU) and as relative percentage of neutralization with respect to the negative control.

### 2.6. Statistical Analysis

The data were analyzed using GraphPad Prism v.8.4.3 software (GraphPad Software, Inc., La Jolla, CA, USA; www.graphpad.com (accessed on 20 January 2022); RRID:SCR_002798). A Kruskal–Wallis test with Dunn’s post-test was performed to compare treatment groups for each vaccine combination. Virus neutralization data by groups was compared by unpaired *t* tests. Significant differences were determined at *p* < 0.05.

## 3. Results

### 3.1. Antigen-Specific IgY Antibodies Were Detected in the Serum of Hyperimmunized Hens

Commercial laying hens were immunized against the SARS-CoV-2 S1 protein, and blood was collected 21 days following the second vaccination for antigen-specific ELISAs. Birds receiving 50 μg doses from all vaccines demonstrated strong antibody responses, which were higher (*p* < 0.05) than negative control titers (Figure 2). At the lower doses (2.5 and 5 μg), only birds receiving vaccine C demonstrated positive antibody responses, which were significantly higher than non-immunized control titers at 2.5 μg. Notably, antibody titers among birds administered vaccine C were not significantly different between doses, whereas antibody titers among birds administered vaccines A and B were significantly higher in the 50 μg groups compared with their respective 2.5 and 5 μg groups (Figure 2A–C).

To test the immunogenicity of combining two vaccines, one group of birds received primary vaccine C followed by a boost with vaccine B. ELISAs were performed 21 days following the second immunization to measure both antigen C- and B-specific antibody responses. The birds receiving 50 μg of vaccine C/B developed strong antigen B- and C-specific IgY titers (Figure 2D,E), which were significantly greater than their respective negative control titers and titers among birds receiving 2.5 and 5 μg doses.

### 3.2. Antigen A-, B-, and C-Specific IgY antibodies Were Detected in the Yolk of Hyperimmunized Hens

Beginning one month following the second immunization, eggs were collected weekly for ELISAs to detect antigen A-, B-, and C-specific IgY titers extracted from egg yolks. Yolk extracts were diluted 1:400, and specific IgY for each antigen was measured by ELISA using antigen A-, B-, or C-covered plates (Figure 3). Antigen-specific IgY titers extracted from yolks collected 6 weeks after the second immunization were significant for all tested antigens in birds receiving 50 μg doses, regardless of which vaccine they were administered. Among the birds administered vaccine A/A, egg yolk-derived antibodies were detectable at the lower immunization doses, but the titers were only significant for the 2.5 μg dose. Notably, the vaccine A/A 50 μg titers were significantly greater than the 2.5 and 5 μg titers. Egg yolk-derived IgY antibodies from birds receiving the lower doses for vaccine B/B were not significantly different from negative control titers. In contrast, egg yolk-derived IgY of hens from the vaccine C/C groups showed significant antibody titers at all doses (Figure 3A–C).

Egg yolk extractions were also analyzed for C- and B-specific antibody responses in hens receiving a combination of vaccine C and B as first and second dose, respectively. B-specific antibody titers in hens administered vaccine C/B were significant (*p* < 0.05) only in hens receiving the 50 μg doses (Figure 3D), which was consistent with the data reported for the hens receiving vaccine B/B. In contrast, C-specific antibody titers in hens receiving vaccine C/B were significant at 2.5, 5, and 50 μg doses, which corresponded to observations noted in hens in the vaccine C/C group (Figure 3E).

Egg yolk extracts showed a mean protein concentration of 2.42 mg/mL with a standard deviation of 0.48, whereas serum protein concentration had a mean of 56.71 mg/mL with a 2.2 standard deviation.

### 3.3. Hyperimmunized Hens Produced Neutralizing Antibodies against SARS-CoV-2

Hens immunized against the SARS-CoV-2 S1 protein were tested for neutralizing antibodies in sera at three weeks after the second immunization and in egg yolk extractions at six weeks following the second immunization. Negative controls included IgY from serum and egg yolk of hens that were either infectious bronchitis virus (IBV)-vaccinated or IBV-naïve. The IBV-naïve birds were tested regularly to confirm the absence of IBV antibodies by ELISA.

Sera and egg yolk IgY extractions from both IBV-vaccinated and IBV-naïve controls did not neutralize the virus. Sera from hens receiving vaccine A/A and B/B neutralized the virus up to 1:20, whereas sera from hens receiving vaccines C/C and C/B neutralized the virus up to 1:40 and 1:80, respectively (Figure 4A). IgY extractions from egg yolk neutralized the virus up to 1:10 among only the hens in the vaccine C/C and C/B groups, and no neutralization was observed among egg yolk samples from hens receiving vaccines A/A and B/B (Figure 4B).

## 4. Discussion

In this study we demonstrate that laying hens hyperimmunized against three different SARS-CoV-2 recombinant spike proteins produced specific antibodies in sera. We further confirmed that the dose and type of antigen administered influenced the levels of antibody titers. Doses were selected based on previous experience and considering a low, medium, and high protein content in each vaccine. In addition, we showed that all vaccines induced neutralizing antibodies against SARS-CoV-2, but that vaccines against the glycosylated S1 protein (C/C) and combination of glycosylated S1 and non-glycosylated RBD of S1 (C/B) led to better neutralization. Combination of C and B antigens was used due to the results obtained in the ELISA from the collected sera showing better responses than for the A antigen. In addition, glycosylated antigen was used as a primer due to its superior response in comparison with the other tested antigens. Specifically, neutralization was observed in all groups, but it was at lower serum concentrations among the hens receiving the full-length glycosylated S1 I protein at least once, whereas virus neutralization by IgY extracted from egg yolk occurred only in the vaccine C/C and C/B groups.

Compared with the negative control group, the full length and tagged S1 protein elicited a higher level (*p* < 0.05) of IgY titers in serum and egg yolk at all doses, whereas immunization with the RBD fragments (A or B; non-tagged) induced antibody titers higher than the negative control (*p* < 0.05) only when immunized at the highest dose of 50 μg. This observation could be explained by the structural difference between antigens. In contrast to antigens A and B, in which the S1 fragments contained only the receptor-binding domain, antigen C comprised the entire extracellular domain of the S1 protein. The increased size and number of potential epitopes may be responsible for the enhanced antigenicity of vaccine C. Another possible explanation could be that antigen C was produced using a eukaryotic HEK293 expression system. In contrast to the prokaryotic expression system used for manufacturing vaccines A and B, HEK293 cells are capable of the post-translational modifications necessary for the correct folding of the protein [19]. Thus, modifications such as phosphorylation or glycosylation of the recombinant protein could potentially enhance immunogenicity [19]. Our data suggest that antibodies generated from vaccine C could therefore be administered at a fraction of the dose necessary for seroconversion to occur. This is crucial when considering potential applications of IgY in prevention strategies against COVID-19.

We also demonstrated that a combination approach using vaccine C followed by vaccine B generated good IgY titers specific for antigen C, which were comparable to the titers observed with vaccine C/C. Similarly, vaccine C/B IgY titers specific for antigen B corresponded to vaccine B/B titers. Importantly, the vaccine C/B combination resulted in neutralizing antibody titers that were at least as good as the titers induced by vaccine C/C. Since the production of proteins using the eukaryotic HEK293 expression system is more costly than using a prokaryotic system, a vaccine combination approach may prove to be a more economical, but equally effective, immunization strategy.

Since the hens had been vaccinated for IBV, a chicken gammacoronavirus, it was important to determine whether cross-neutralization could be observed between samples from IBV-vaccinated hens and SARS-CoV-2. Virus neutralization was not observed by IBV-vaccinated hens, which suggests that IBV antibodies do not cross-react with SARS-CoV-2. Although unsurprising given the enormous genetic variability of coronaviruses, this observation underscores the importance of ensuring that the spike protein used for vaccination is genetically similar to the spike protein of the circulating virus.

As expected, SARS-CoV-2-specific IgY titers were also detected in egg yolk. As shown in our results, IgY titers in yolk were lower than in serum. These outcomes might be related to the efficiency of our yolk antibody extraction method, which is based only on ultrapure water and centrifugation. The finding of a reduced egg yolk extract protein concentration compared to serum protein concentration further supports this claim. This extraction inefficiency might also explain the lower neutralization activity demonstrated by the yolk IgY in the plaque neutralization test. The egg yolk is a complex structure composed of proteins, lipids, vitamins, and minerals. The main components of the egg yolk are lipids in ~65% of dry matter, and the lipid-to-protein ratio is about 2:1 [20]. In addition, lipids in the yolk are exclusively associated with lipoprotein assemblies, i.e., triglycerides, phospholipids, and cholesterol [20]. This intricate lipid association with proteins can complicate protein extraction from the yolk if the lipids are not diluted appropriately, which might have been the case with our IgY extraction method. Anecdotally, we have worked with a private company upscaling the hyperimmunization and egg production systems. They have used a polyethylene glycol precipitation to extract IgY from yolks with better results (Michelle Hawkins, CAMAS, personal communication). The polyethylene glycol method has been tested by several researchers [12,21] and seems to be a better method to address the issue of fats present in the egg yolk. This methodology is crucial for pure antibody extractions and formulation of preventative products, such as local use sprays containing IgY.

A recent publication demonstrated that IgY extracted from egg yolk of hens immunized with the RBD of SARS-CoV-2 could prevent entry and replication of SARS-CoV-2 in vitro [22]. These results strengthen the case for administering egg yolk-derived IgY to reduce SARS-CoV-2 infection. In this study the neutralizing antibody titer was relatively low and for commercial success will require further optimization to increase the titer. As an example, the egg yolk extraction method could be optimized to improve IgY extraction efficacy. The most likely application of such an approach involves a spray to coat the nose and throat of at-risk humans to reduce the risk of SARS-CoV-2 infection.

For some of the measured responses, differences were minor; this might be explained by the size limitations of our BSL2 facilities. While a sample size of 30 hens per group would have given a statistical power of 80% and 5% significance, we were only able to house 17 to 18 birds per group, limiting our statistical power.

The IgY neutralizing effects measured in this study were performed using the USA-WA1/2020. Like many other RNA viruses including the coronaviruses, SARS-CoV-2 is prone to frequent mutations [23,24]. Thus far, thousands of mutations have been observed at the nucleotide level in the SARS-CoV-2 genome, indicating that SARS-CoV-2 has already become heterogeneous [23]. The emergence of more prevalent new variants has already resulted in reduced efficacy of current vaccines, so it is critical to closely monitor the evolution of SARS-CoV-2 to ensure that prophylactic antibody administration targeted toward the SARS-CoV-2 RBD retains sufficient neutralizing activity.

## Figures and Tables

**Figure 1 viruses-14-01510-f001:**
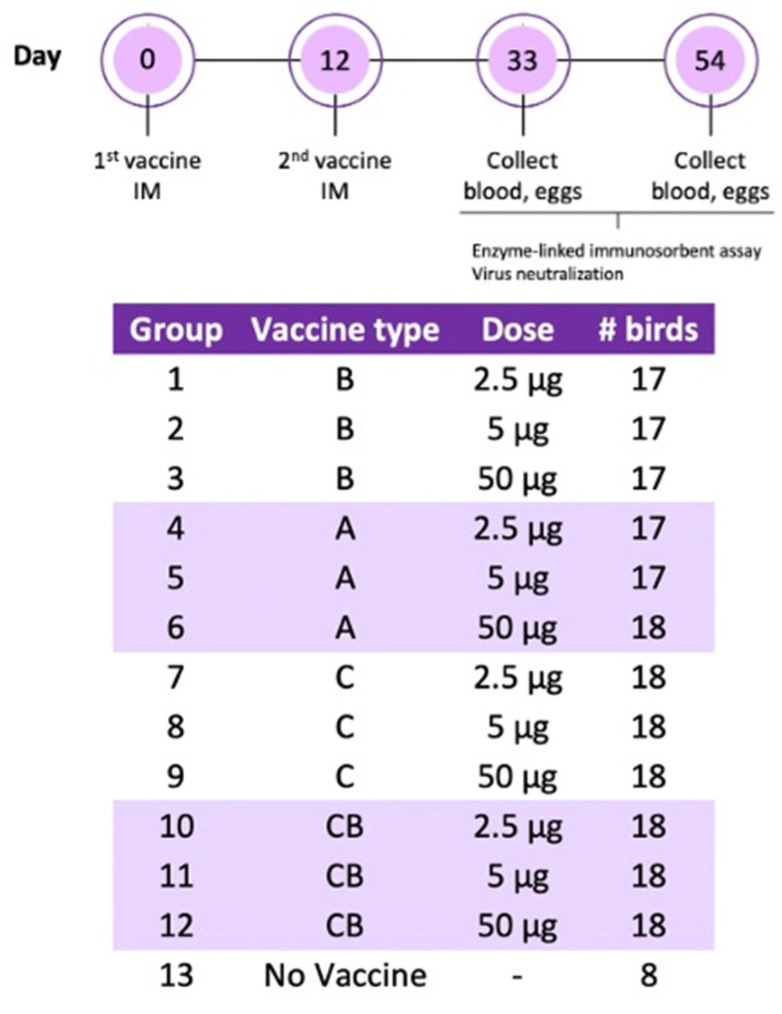
Schematic of the experimental design.

**Figure 2 viruses-14-01510-f002:**
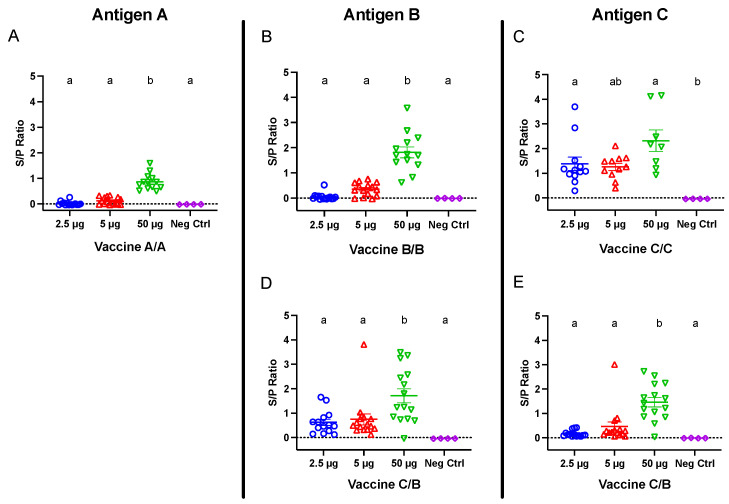
IgY titers in serum collected 21 days following the second vaccination for groups receiving (**A**) vaccine A/A, (**B**) vaccine B/B, (**C**) vaccine C/C, and (**D**,**E**) vaccine C/B combinations. Hens were inoculated twelve days apart. Antigen A-, B-, and C-coated plates were used for panels (**A**), (**B**,**D**), and (**C**,**E**), respectively. Error bars indicate the mean and standard error. Means without a common letter (a, b) are significantly different (*p* < 0.05).

**Figure 3 viruses-14-01510-f003:**
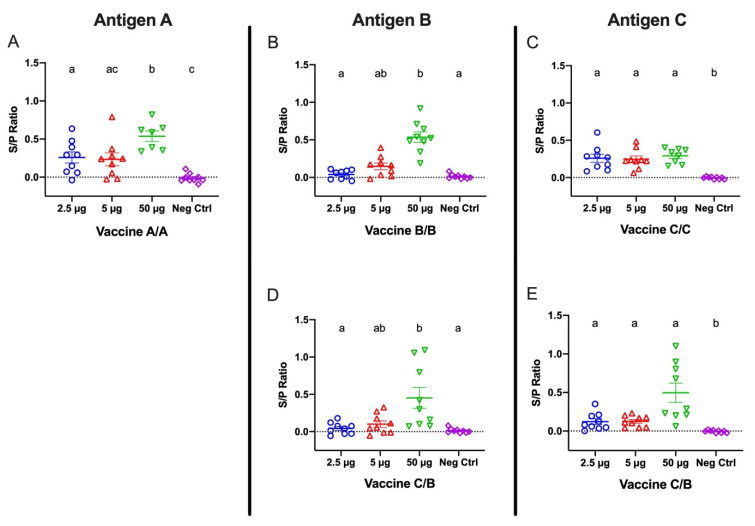
IgY titers from egg yolk collected six weeks following the second vaccination for groups receiving (**A**) vaccine A/A, (**B**) vaccine B/B, (**C**) vaccine C/C, and (**D**,**E**) vaccine C/B combinations. Hens were inoculated twelve days apart. Antigen A-, B-, and C-coated plates were used for panels (**A**), (**B**,**D**) and (**C**,**E**), respectively. Error bars indicate the mean and standard error. Means without a common letter (a–c) are significantly different (*p* < 0.05).

**Figure 4 viruses-14-01510-f004:**
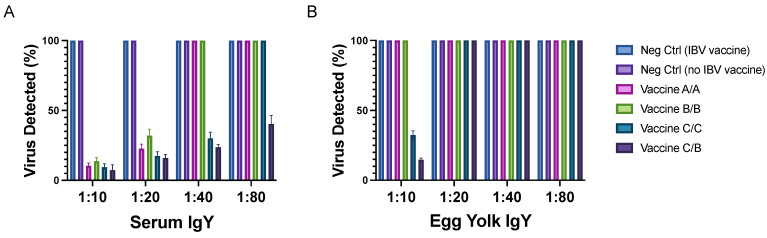
A plaque reduction neutralization assay (PRNA) was performed to detect SARS-CoV-2-specific neutralizing antibodies in (**A**) sera and (**B**) egg yolk IgY extractions from hens receiving 50 μg of vaccine A/A, B/B, C/C, and C/B. Controls included negative control hens that were either vaccinated or not vaccinated for IBV. Sera was collected at three weeks after the second vaccination, and egg yolk was harvested beginning at six weeks after the second vaccination. Viral titer of SARS-CoV-2 was determined by counting the number of plaques and represented as relative percentage of neutralization with respect to the negative control. The PRNA was performed in triplicate.

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
