# Peer review of "Hyperimmunized Chickens Produce Neutralizing Antibodies against SARS-CoV-2"

_viruses, 2022, doi:10.3390/v14071510_

Round 1

Reviewer 1 Report

The presence of antibodies in chicken egg yolk has been well described. The authors present timely work showing a straightforward method for the generation and harvest of antibodies in chicken egg yolk for possible therapeutic use in humans. In my opinion, the construction of a biologically relevant recombinant vaccine containing all or only targeted receptor binding domains of the spike protein for any coronavirus is not trivial yet, the authors did not present the design or validation of the recombinant vaccines used in the study. It would be nice to present additional validation of the vaccines A, B and C prior to use in in vivo trials.  The fact that recombinant vaccines were created such that the vaccine could induce biologically functional antibodies in chickens was fascinating.  I think the most interesting part of this manuscript is not presented in sufficient detail.

Specific comments:

Why does Figure 4 appear first?

The Figures 1 and 2 are clearly presented however, I am left to wonder about the biological relevance of presented data. I would specifically suggest changing the Y axis values in Fig 1, panel Antigen A to match those of the other panels.

Line 276: The authors refer to “good levels” in sera. I am not sure what good levels are. Would be nice to add a specific number to in this sentence.

Figure 3. I would you ask the authors to consider changing the Y-axis label of “virus detected” to something pertaining to plaques observed or reduction of viral plaques.  It would be nice if the discussion contained something about the assay and the forced interaction between virus and antibody prior to plating. This is something that clearly would not occur naturally. In addition, was egg yolk IgY tested at a 1:5 dilution? Seems like that would be a logical step to increase the efficacy of the antibodies.

Author Response

Reviewer #1:

The presence of antibodies in chicken egg yolk has been well described. The authors present timely work showing a straightforward method for the generation and harvest of antibodies in chicken egg yolk for possible therapeutic use in humans. In my opinion, the construction of a biologically relevant recombinant vaccine containing all or only targeted receptor binding domains of the spike protein for any coronavirus is not trivial yet, the authors did not present the design or validation of the recombinant vaccines used in the study. It would be nice to present additional validation of the vaccines A, B and C prior to use in in vivo trials.  The fact that recombinant vaccines were created such that the vaccine could induce biologically functional antibodies in chickens was fascinating.  I think the most interesting part of this manuscript is not presented in sufficient detail.

We appreciate the reviewer’s comments. The purpose of this study was to generate IgY in hens and prove the functionality of those antibodies through VN tests. Exploring vaccines using artificially created sequences of the receptor binding domains is a fascinating topic unfortunately, this was out of the scope of this investigation.   

Specific comments:

Why does Figure 4 appear first?

Figure numbering was changed to reflect the order of appearance in the manuscript.

The Figures 1 and 2 are clearly presented however, I am left to wonder about the biological relevance of presented data. I would specifically suggest changing the Y axis values in Fig 1, panel Antigen A to match those of the other panels.

The biological relevance of the data presented in figure 2 (previously #1) is that hens responded adequately in a dose dependent fashion to the inoculation of each of the vaccines and combinations, while responses were different for the different constructs used. This was reported in the results and the discussion section of the manuscript. The Y axis on figure 2A was changed to match the rest of the graphs in the figure.

Line 276: The authors refer to “good levels” in sera. I am not sure what good levels are. Would be nice to add a specific number to in this sentence.

The sentence was changed for accuracy in line 273

Figure 3. I would you ask the authors to consider changing the Y-axis label of “virus detected” to something pertaining to plaques observed or reduction of viral plaques.  It would be nice if the discussion contained something about the assay and the forced interaction between virus and antibody prior to plating. This is something that clearly would not occur naturally. In addition, was egg yolk IgY tested at a 1:5 dilution? Seems like that would be a logical step to increase the efficacy of the antibodies.

The viral titer of SARS-CoV-2 in the assay was determined by counting the number of plaques and represented as a relative percentage of neutralization with respect to the negative control. We believe that this representation is comprehensive because it considers plaque formation and is corrected by the negative control. We believe this is a good test to determine the neutralization capabilities of the antibodies generated. Unfortunately, we did not test the egg yolk at a 1:5 dilution because we wanted to be comparable with the assay performed with the sera.      

Reviewer 2 Report

Brief explanation/description may be needed to provide why 2.5, 5 and 50 dosde range were applied

Brief explanation may be needed why C&B prime-booster comnbination was applied (there are many other possible combinations)

thecstat power of 17 or 18 birds are very limited,  the significance of your results are not convincing enough, acknowledging animal welfare perspectives at least a scientific publication based - more detailed- stat analysis should be added togetzher with a descrioption that  a given test item examined on 17-18 birds in repeated studies may give a results in a reasonable narrow range. 

Technically a demonstartion would be needed to add that a ELISA against IBV could be a suitable test system for SARS-CoV-2 antibody screening (previous sci pub data, comparative anayslis, that IBS S and SARS-CoV-2 S gebnes are closely related, etc)

A brief perspective (citations) to be added that chicken basd antibodies can be properly used in humans for therapeutic purposes without any immune side-effects

Author Response

Reviewer #2:

Brief explanation/description may be needed to provide why 2.5, 5 and 50 dose range were applied

A brief explanation was added in the discussion section, lines 275 to 276.

Brief explanation may be needed why C&B prime-booster combination was applied (there are many other possible combinations)

A brief explanation was added in the discussion section lines 279 to 282. 

The stat power of 17 or 18 birds are very limited, the significance of your results are not convincing enough, acknowledging animal welfare perspectives at least a scientific publication based - more detailed- stat analysis should be added togetzher with a descrioption that  a given test item examined on 17-18 birds in repeated studies may give a results in a reasonable narrow range.

We understand the reviewer’s concern. In our experience groups of 17 to 18 hens is a good number to draw conclusions about vaccine responses and population immunity. Historically and for regulatory purposes, 10 blood samples are collected to determine vaccine takes in poultry flocks. Our experiments usually use between 10 to 15 birds to study viral infection, immune responses, viral clearance and disease characterization. In addition, 17 to 18 was the best number we could house in our BSL2 facilities.

Assuming a standard deviation of 1.37 units, the study would require a sample size of: 30 for each group, to achieve a power of 80% and a level of significance of 5% (two sided), for detecting a true difference in means between the test and the reference group of 0.98 (i.e. 26.54 - 25.56) units. Unfortunately, this is not possible.

Technically a demonstartion would be needed to add that a ELISA against IBV could be a suitable test system for SARS-CoV-2 antibody screening (previous sci pub data, comparative anayslis, that IBS S and SARS-CoV-2 S gebnes are closely related, etc)

The ELISA used to detect antibodies specific to the used antigens was developed “in-house”. We only used the anti-chicken antibody and TMB from the IBV IDEXX ELISA. This information is reported in the materials and methods section line 144 to 166.  

A brief perspective (citations) to be added that chicken basd antibodies can be properly used in humans for therapeutic purposes without any immune side-effects

Information and citations regarding application and safety in the use of IgY in humans is included in the manuscript under the introduction section, lines 79 to 85

Reviewer 3 Report

  • On what basis, the construct for Vaccine C was selected for use in this study?
  • Did authors perform the sequence analysis for similarity and diversity between USA and Wuhan isolate? If yes, then they should mention in the text to make it clear that the reason for selection of Full S1 and RBD.
  • What was the percent sequence diversity between S1 of SARS-CoV-2 and IBV?
  • Why the antibody titer was not significantly different in Vaccine C at different doses concentration? But the fig 1 -C shows the significant differences in antibody titers in the serum sample.
  • In figure 3 b, only vaccine C/C and C/B combinations showed the Virus neutralization at 50 micrograms with a 1:10 ratio, but it is  not mentioned other concentrations. Additionally, the authors did not mention the explanation of the low neutralizing activity of the egg yolk antibodies. Authors may also evaluate the combinations of A/C, and A/B and present the data to make it more clear and enhance the impact of this MS.
  • The purity of the extracted antibodies needs to be determined by SDS in both serum and egg yolk.
  • Western blot is needed to determine the specificity of antibodies raised against each recombinant protein.
  • No ethical approval Number was given in the text for the chicken experiment.
  • Authors should also describe the effect of doses on antibody titer in serum, egg yolk, and level of virus neutralization at different doses for all the vaccines evaluated. Additionally, the authors should explain, why the level of virus neutralization in serum is more potent than the IgY egg yolk (Usually, it must be similar).
  • Why did only Vaccine C give significant virus neutralization? Is there any link between position and number of amino acids differences? As well as the size of the target (Full S1/ RBD) or any effect of glycosylation?
  • Is there any significant role of glycosylated/non-glycosylated protein on the trigger of immunogenic response and level of antibody titer in both serum and egg yolk?
  • In the discussion part, the authors have mentioned that the structural differences could be the reason for high antibody titer as seen high in Vaccine C, but if this is the main reason, then they should also get the similar differences in other vaccines also with lower antibody level or very low, but in the figure, it is not like that, it can be explained in the discussion.
  • If the structural differences are the main reason, then authors can also perform the epitope mapping and present one figure, and the structural differences, as well as the immunogenic part and binding region, can be highlighted in the figure for all the tested vaccines. This data will increase the impact of this paper.
  • Authors may also show how many amino acids with the immunogenic region were different in Vaccine C than in other vaccine constructs.
  • Authors have discussed the eukaryotic/prokaryotic system differences in the level of antibodies, but I think that the purification of antibodies from egg yolk is not perfect, they may use the latest kit for purification to get more and better-purified antibodies.
  • Authors could have used the recombinant protein only from the USA isolate to avoid any genomic diversity, structural differences, and level of antibodies in serum and egg yolk which further affects the level of virus neutralization. Is there any study done to support that, Wuhan recombinant protein will work against USA virus isolate?

Author Response

Reviewer #3:

On what basis, the construct for Vaccine C was selected for use in this study?

The construct on vaccine C seemed like a good method to test the effect of glycosylation in the immune response of the hyperimmunized hens. In addition, represents a full S1 compared with A and B that represent the receptor binding domain. This is a commercially available construct.

Did authors perform the sequence analysis for similarity and diversity between USA and Wuhan isolate? If yes, then they should mention in the text to make it clear that the reason for selection of Full S1 and RBD.

This was not done. The reasons for the antigen selection are explained above.

What was the percent sequence diversity between S1 of SARS-CoV-2 and IBV?

The IBV S1 HVR is between 37 to 45% similar to the same region of SARS-CoV-2.

An IBV complete genome is ~70% similar than the SARS-CoV-2 genome

Why the antibody titer was not significantly different in Vaccine C at different doses concentration? But the fig 1 -C shows the significant differences in antibody titers in the serum sample.

While a numerical difference was seen in the vaccine C group (serum) used at 50ug the antibody production at different doses was not significantly different in sera or in yolk

In figure 3 b, only vaccine C/C and C/B combinations showed the Virus neutralization at 50 micrograms with a 1:10 ratio, but it is not mentioned other concentrations. Additionally, the authors did not mention the explanation of the low neutralizing activity of the egg yolk antibodies. Authors may also evaluate the combinations of A/C, and A/B and present the data to make it more clear and enhance the impact of this MS.

Indeed, we only saw neutralization elicited by the egg yolk IgY at the 1:10 dilution in groups vaccine C/C and C/B, the other dilutions did not show neutralization, as shown in figure 4.

The lower neutralization efficiency of the IgY obtained from yolk is explained in the discussion section lines 318 to 324.

Since hens responded considerably lower to antigen A (results from IgY in serum), we decided to focus on the antigens that responded better.

The purity of the extracted antibodies needs to be determined by SDS in both serum and egg yolk.

Western blot is needed to determine the specificity of antibodies raised against each recombinant protein.

We agree with the reviewer comments, since this was a preliminary investigation to test the feasibility of the system, we focused on the antibody immune responses and neutralization caused by these antibodies.

No ethical approval Number was given in the text for the chicken experiment.

The IACUC approval number in this journal is published as a separate statement. The authorization number is in lines 368 and 369.

Authors should also describe the effect of doses on antibody titer in serum, egg yolk, and level of virus neutralization at different doses for all the vaccines evaluated. Additionally, the authors should explain, why the level of virus neutralization in serum is more potent than the IgY egg yolk (Usually, it must be similar).

Results and discussion sections were revised to include the suggested topics. Question/comment #5 addresses the neutralization differences between serum and yolk extracts.

Why did only Vaccine C give significant virus neutralization? Is there any link between position and number of amino acids differences? As well as the size of the target (Full S1/ RBD) or any effect of glycosylation?

These are interesting and valid comments but they are out of the scope of this investigation.  

Is there any significant role of glycosylated/non-glycosylated protein on the trigger of immunogenic response and level of antibody titer in both serum and egg yolk?

This is a very interesting assumption but, is out of the scope of this investigation.

In the discussion part, the authors have mentioned that the structural differences could be the reason for high antibody titer as seen high in Vaccine C, but if this is the main reason, then they should also get the similar differences in other vaccines also with lower antibody level or very low, but in the figure, it is not like that, it can be explained in the discussion.

The discussion section mentions other hypothetical reasons of why better results were obtained when antigen C was used as an antigen e.g. expression system differences and glycosylation. At this point all reasons are potentially valid and expressed in the discussion section. Discussion lines 282 to 302.  

If the structural differences are the main reason, then authors can also perform the epitope mapping and present one figure, and the structural differences, as well as the immunogenic part and binding region, can be highlighted in the figure for all the tested vaccines. This data will increase the impact of this paper. Authors may also show how many amino acids with the immunogenic region were different in Vaccine C than in other vaccine constructs.

This project was a preliminary investigation to test the feasibility of producing IgY in chickens, capable of neutralizing SARS CoV-2. For that we focused on the antibody immune responses and neutralization caused by these antibodies. Even though the reviewer suggestions are interesting and valid, they are out of the scope of this investigation.

Authors have discussed the eukaryotic/prokaryotic system differences in the level of antibodies, but I think that the purification of antibodies from egg yolk is not perfect, they may use the latest kit for purification to get more and better-purified antibodies.

We agree with the reviewer comments, the limitations of the purification system used were disclosed in the manuscript  

Authors could have used the recombinant protein only from the USA isolate to avoid any genomic diversity, structural differences, and level of antibodies in serum and egg yolk which further affects the level of virus neutralization. Is there any study done to support that, Wuhan recombinant protein will work against USA virus isolate?

Those are interesting suggestions, at the time when the experiment was done, the available proteins were the ones used in this experiment. The authors are not aware of publications using Wuhan recombinant proteins against USA virus isolates adding importance to the work presented in this manuscript.    

Round 2

Reviewer 2 Report

Brief explanation/description may be needed to provide why 2.5, 5 and 50 dose range were applied

A brief explanation was added in the discussion section, lines 275 to 276.- ACCEPTED

Brief explanation may be needed why C&B prime-booster combination was applied (there are many other possible combinations)

A brief explanation was added in the discussion section lines 279 to 282. - ACCEPTED

The stat power of 17 or 18 birds are very limited, the significance of your results are not convincing enough, acknowledging animal welfare perspectives at least a scientific publication based - more detailed- stat analysis should be added togetzher with a descrioption that  a given test item examined on 17-18 birds in repeated studies may give a results in a reasonable narrow range.

We understand the reviewer’s concern. In our experience groups of 17 to 18 hens is a good number to draw conclusions about vaccine responses and population immunity. Historically and for regulatory purposes, 10 blood samples are collected to determine vaccine takes in poultry flocks. Our experiments usually use between 10 to 15 birds to study viral infection, immune responses, viral clearance and disease characterization. In addition, 17 to 18 was the best number we could house in our BSL2 facilities.

Assuming a standard deviation of 1.37 units, the study would require a sample size of: 30 for each group, to achieve a power of 80% and a level of significance of 5% (two sided), for detecting a true difference in means between the test and the reference group of 0.98 (i.e. 26.54 - 25.56) units. Unfortunately, this is not possible.

Understanding the capacity limitations of BSL2, your calculation clearly indicates the expected 80% stat popwer and 5% significance limit can be achieved by using 25-26 birds. You should indicate the stat power and significance limit when discussing your results, because otherwise the readers assume the "usual" 80% power and 5% significanxce limti, but here is not the case.

Technically a demonstartion would be needed to add that a ELISA against IBV could be a suitable test system for SARS-CoV-2 antibody screening (previous sci pub data, comparative anayslis, that IBS S and SARS-CoV-2 S gebnes are closely related, etc)

The ELISA used to detect antibodies specific to the used antigens was developed “in-house”. We only used the anti-chicken antibody and TMB from the IBV IDEXX ELISA. This information is reported in the materials and methods section line 144 to 166.  - ACCEPTED

A brief perspective (citations) to be added that chicken basd antibodies can be properly used in humans for therapeutic purposes without any immune side-effects

Information and citations regarding application and safety in the use of IgY in humans is included in the manuscript under the introduction section, lines 79 to 85 - ACCEPTED

Author Response

We understand the reviewer's concern. We addressed this issue in the discussion section lines 345 to 348.

Reviewer 3 Report

SDS PAGE AND WESTERN BLOTTING IS MUST.

Author Response

We appreciate the reviewer comments. We believe that all the data and information shared in this manuscript reflects the potential of this strategy to provide alternative preventative strategies for COVID infections. As a pilot experiment it has some flaws and those have been shared and discussed in this manuscript.